# TRANSFORM2ACT: LEARNING A TRANSFORM-AND-CONTROL POLICY FOR EFFICIENT AGENT DESIGN

**Ye Yuan[1], Yuda Song[1], Zhengyi Luo[1], Wen Sun[2], Kris M. Kitani[1]**
[1]Carnegie Mellon University,    [2]Cornell University
{yyuan2,yudas,zluo2,kkitani}@cs.cmu.edu, ws455@cornell.edu

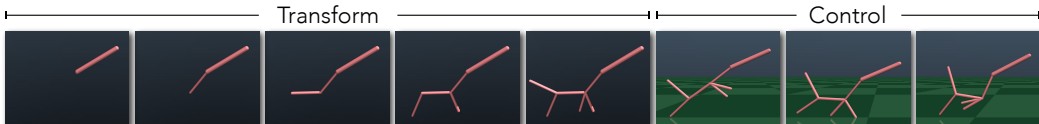

Figure 1: Transform2Act learns a transform-and-control policy that first applies transform actions to design an agent and then controls the designed agent to interact with the environment. The giraffe-like agent obtained by Transform2Act can run extremely fast and remain stable (see video).

## ABSTRACT

An agent's functionality is largely determined by its design, i.e., skeletal structure and joint attributes (e.g., length, size, strength). However, finding the optimal agent design for a given function is extremely challenging since the problem is inherently combinatorial and the design space is prohibitively large. Additionally, it can be costly to evaluate each candidate design which requires solving for its optimal controller. To tackle these problems, our key idea is to incorporate the design procedure of an agent into its decision-making process. Specifically, we learn a conditional policy that, in an episode, first applies a sequence of transform actions to modify an agent's skeletal structure and joint attributes, and then applies control actions under the new design. To handle a variable number of joints across designs, we use a graph-based policy where each graph node represents a joint and uses message passing with its neighbors to output joint-specific actions. Using policy gradient methods, our approach enables joint optimization of agent design and control as well as experience sharing across different designs, which improves sample efficiency substantially. Experiments show that our approach, Transform2Act, outperforms prior methods significantly in terms of convergence speed and final performance. Notably, Transform2Act can automatically discover plausible designs similar to giraffes, squids, and spiders. Code and videos are available at https://sites.google.com/view/transform2act.

## 1 INTRODUCTION

Automatic and efficient design of robotic agents in simulation holds great promise in complementing and guiding the traditional physical robot design process (Ha et al., 2017; 2018), which can be laborious and time-consuming. In this paper, we consider a setting where we optimize both the skeletal structure and joint attributes (e.g., bone length, size, and motor strength) of an agent to maximize its performance on a given task. This is a very challenging problem for two main reasons. First, the design space, i.e., all possible skeletal structures and their joint attributes, is prohibitively vast and combinatorial, which also makes applying gradient-based continuous optimization methods difficult. Second, the problem is inherently bi-level: (1) we need to search an immensely large design space and (2) the evaluation of each candidate design entails solving a computationally expensive inner optimization to find its optimal controller. Prior work typically uses evolutionary search (ES) algorithms for combinatorial design optimization (Sims, 1994; Wang et al., 2019). During each iteration, ES-based methods maintain a large population of agents with various designs where each agent learns to perform the task independently. When the learning ends, agents with the worst performances are eliminated while surviving agents produce child agents with random mutation to maintain the size of the population. ES-based methods have low sample efficiency since agents in the population do not share their experiences and many samples are wasted on eliminated agents. Fur-

thermore, zeroth-order optimization methods such as ES are known to be sample-inefficient when the optimization search space (i.e., design space) is high-dimensional (Vemula et al., 2019).

In light of the above challenges, we take a new approach to agent design optimization by incorporating the design procedure of an agent into its decision-making process. Specifically, we learn a transform-and-control policy, called Transform2Act, that first designs an agent and then controls the designed agent. In an episode, we divide the agent's behavior into two consecutive stages, a transform stage and an execution stage, on which the policy is conditioned. In the transform stage, the agent applies a sequence of transform actions to modify its skeletal structure and joint attributes without interacting with the environment. In the execution stage, the agent assumes the design resulting from the transform stage and applies motor control actions to interact with the environment and receives rewards. Since the policy needs to be used across designs with a variable number of joints, we adopt graph neural networks (GNNs) as the policy's main network architecture. Each graph node in the GNNs represents a joint and uses message passing with its neighbors to output joint-specific actions. While GNNs can improve the generalizability of learned control policies across different skeletons through weight sharing, they can also limit the specialization of each joint's design since similar joints tend to output similar transform actions due to the weight sharing in GNNs. To tackle this problem, we propose to attach a joint-specialized multilayer perceptron (JSMLP) on top of the GNNs in the policy. The JSMLP uses different sets of weights for each joint, which allows more flexible transform actions to enable asymmetric designs with more specialized joint functions.

The proposed Transform2Act policy jointly optimizes its transform and control actions via policy gradient methods, where the training batch for each iteration includes samples collected under various designs. Unlike zeroth-order optimization methods (e.g., ES) that do not use a policy to change designs but instead mutate designs randomly, our approach stores information about the goodness of a design into our transform policy and uses it to select designs to be tested in the execution stage. The use of GNNs further allows the information stored in the policy to be shared across joints, enabling better experience sharing and prediction generalization for both design and control. Furthermore, in contrast to ES-based methods that do not share training samples across designs in a generation, our approach uses all the samples from all designs to train our policy, which improves sample efficiency.

The main contributions of this paper are: (1) We propose a transform-and-control paradigm that formulates design optimization as learning a conditional policy, which can be solved using the rich tools of RL. (2) Our GNN-based conditional policy enables joint optimization of design and control as well as experience sharing across all designs, which improves sample efficiency substantially. (3) We further enhance the GNNs by proposing a joint-specialized MLP to balance the generalization and specialization abilities of our policy. (4) Experiments show that our approach outperforms previous methods significantly in terms of convergence speed and final performance, and is also able to discover familiar designs similar to giraffes, squids, and spiders.

## 2 RELATED WORK

**Continuous Design Optimization.** Considerable research has examined optimizing an agent's continuous design parameters without changing its skeletal structure. For instance, Baykal & Alterovitz (2017) introduce a simulated annealing-based optimization framework for designing piecewise cylindrical robots. Alternatively, trajectory optimization and the implicit function theorem have been used to adapt the design of legged robots (Ha et al., 2017; 2018; Desai et al., 2018). Recently, deep RL has become a popular approach for design optimization. Chen et al. (2020) model robot hardware as part of the policy using computational graphs. Luck et al. (2020) learn a design-conditioned value function and optimize design via CMA-ES. Ha (2019) uses a population-based policy gradient method for design optimization. Schaff et al. (2019) employs RL and evolutionary strategies to maintain a distribution over design parameters. Another line of work (Yu et al., 2018; Exarchos et al., 2020; Jiang et al., 2021) uses RL to find the robot parameters that best fit an incoming domain. Unlike the above works, our approach can optimize the skeletal structure of an agent in addition to its continuous design parameters.

**Combinatorial Design Optimization.** Towards jointly optimizing the skeletal structure and design parameters, the seminal work by Sims (1994) uses evolutionary search (ES) to optimize the design and control of 3D blocks. Extending this method, Cheney et al. (2014; 2018) adopt oscillating 3D voxels as building blocks to reduce the search space. Desai et al. (2017) use human-in-the-loop tree search to optimize the design of modular robots. Wang et al. (2019) propose an evolutionary

graph search method that uses GNNs to enable weight sharing between an agent and its offspring. Recently, Hejna et al. (2021) employ an information-theoretic objective to evolve task-agnostic agent designs. Due to the combinatorial nature of skeletal structure optimization, most prior works use ES-based optimization frameworks which can be sample-inefficient since agents with various designs in the population learn independently. In contrast, we learn a transform-and-control policy using samples collected from all designs, which improves sample efficiency significantly.

**GNN-based Control.** Graph neural networks (GNNs) (Scarselli et al., 2008; Bruna et al., 2013; Kipf & Welling, 2016) are a class of models that use message passing to aggregate and extract features from a graph. GNN-based control policies have been shown to greatly improve the generalizability of learned controllers across agents with different skeletons (Wang et al., 2018; 2019; Huang et al., 2020). Along this line, Pathak et al. (2019) use a GNN-based policy to control self-assembling modular robots to perform tasks. Recently, Kurin et al. (2021) show that GNNs can hinder learning in incompatible multitask RL and propose to use attention mechanisms instead. In this paper, we also study the lack of per-joint specialization caused by GNNs due to weight sharing, and we propose a remedy, joint-specialized MLP, to improve the specialization of GNN-based policies.

## 3 BACKGROUND

**Reinforcement Learning.** Given an agent interacting with an episodic environment, reinforcement learning (RL) formulates the problem as a Markov Decision Process (MDP) defined by a tuple $\mathcal{M} = (\mathcal{S}, \mathcal{A}, \mathcal{T}, R, \gamma)$ of state space, action space, transition dynamics, a reward function, and a discount factor. The agent's behavior is controlled by a policy $\pi(a_t|s_t)$, which models the probability of choosing an action $a_t \in \mathcal{A}$ given the current state $s_t \in \mathcal{S}$. Starting from some initial state $s_0$, the agent iteratively samples an action $a_t$ from the policy $\pi$ and the environment generates the next state $s_{t+1}$ based on the transition dynamics $\mathcal{T}(s_{t+1}|s_t, a_t)$ and also assigns a reward $r_t$ to the agent. The goal of RL is to learn an optimal policy $\pi^*$ that maximizes the expected total discounted reward received by the agent: $J(\pi) = \mathbb{E}_\pi \left[ \sum_{t=0}^H \gamma^t r_t \right]$, where $H$ is the variable time horizon. In this paper, we use a standard policy gradient method, PPO (Schulman et al., 2017), to optimize our policy with both transform and control actions. PPO is particularly suitable for our approach since it has a KL regularization between current and old policies, which can prevent large changes to the transform actions and the resulting design in each optimization step, thus avoiding catastrophic failure.

**Design Optimization.** An agent's design $D \in \mathcal{D}$ plays an important role in its functionality. In our setting, the design $D$ includes both the agent's skeletal structure and joint-specific attributes (e.g., bone length, size, and motor strength). To account for changes in design, we now consider a more general transition dynamics $\mathcal{T}(s_{t+1}|s_t, a_t, D)$ conditioned on design $D$. The total expected reward is also now a function of design $D$: $J(\pi, D) = \mathbb{E}_{\pi, D} \left[ \sum_{t=0}^H \gamma^t r_t \right]$. One main difficulty of design optimization arises from its bi-level nature, i.e., we need to search over a large design space and solve for the optimal policy under each candidate design for evaluation. Formally, the bi-level optimization is defined as:

$$D^* = \arg\max_D J(\pi_D, D) \tag{1}$$

$$\text{subject to} \quad \pi_D = \arg\max_\pi J(\pi, D) \tag{2}$$

The inner optimization described by Equation (2) typically requires RL, which is computationally expensive and may take up to several days depending on the task. Additionally, the design space $\mathcal{D}$ is extremely large and combinatorial due to a vast number of possible skeletal structures. To tackle these problems, in Section 4 we will introduce a new transform-and-control paradigm that formulates design optimization as learning a conditional policy to both design and control the agent. It uses first-order policy optimization via policy gradient methods and enables experience sharing across designs, which improves sample efficiency significantly.

**Graph Neural Networks.** Since our goal is to learn a policy to dynamically change an agent's design, we need a network architecture that can deal with variable input sizes across different skeletal structures. As skeletons can naturally be represented as graphs, graph neural networks (GNNs) (Scarselli et al., 2008; Bruna et al., 2013; Kipf & Welling, 2016) are ideal for our use case.

We denote a graph as $G = (V, E, A)$ where $u \in V$ and $e \in E$ are nodes and edges respectively, and each node $u$ also includes an input feature $x_u \in A$. A GNN uses multiple GNN layers to extract and

aggregate features from the graph $G$ through message passing. For the $i$-th of $N$ GNN layers, the message passing process can be written as:

$$m_u^i = M(h_u^{i-1}), \tag{3}$$

$$c_u^i = C(\{m_v^i \mid \forall v \in \mathcal{N}(u)\}), \tag{4}$$

$$h_u^t = U(h_u^{i-1}, c_u^i), \tag{5}$$

where a message sending module $M$ first computes a message $m_u^i$ for each node $u$ from the hidden features $h_u^{i-1}$ ($h_u^0 = x_u$) of the previous layer. Each node's message is then sent to neighboring nodes $\mathcal{N}(u)$, and an aggregation module $C$ summarizes the messages received by every node and outputs a combined message $c_u^i$. Finally, a state update module $U$ updates the hidden state $h_u^i$ of each node using $c_u^i$ and previous hidden states $h_u^{i-1}$. After $N$ GNN layers, an output module $P$ is often used to regress the final hidden features $h_u^N$ to desired outputs $y_u = P(h_u^N)$. Different designs of modules $M, C, U, P$ lead to many variants of GNNs (Wu et al., 2020). We use the following equation to summarize GNNs' operations:

$$y_u = \text{GNN}(u, A; V, E) \tag{6}$$

where $\text{GNN}(u, \cdot)$ is used to denote the output for node $u$.

## 4 TRANSFORM2ACT: A TRANSFORM-AND-CONTROL POLICY

To tackle the challenges in design optimization, our key approach is to incorporate the design procedure of an agent into its decision-making process. In each episode, the agent's behavior is separated into two consecutive stages: (1) **Transform Stage**, where the agent applies transform actions to modify its design, including skeletal structure and joint attributes, *without* interacting with the environment; (2) **Execution Stage**, where the agent assumes the new transformed design and applies motor control actions to interact with the environment. In both stages, the agent is governed by a conditional policy, called *Transform2Act*, that selects transform or control actions depending on which stage the agent is in. In the transform stage, no environment reward is assigned to the agent, but the agent will see future rewards accrued in the execution stage under the transformed design, which provide learning signals for the transform actions. By training the policy with PPO (Schulman et al., 2017), the transform and control actions are optimized jointly to improve the agent's performance for the given task. An overview of our method is provided in Figure 2. We also outline our approach in Algorithm 1. In the following, we first introduce the preliminaries before describing the details of the proposed Transform2Act policy and the two stages.

**Design Representation.** To represent various skeletal structures, we denote an agent design as a graph $D_t = (V_t, E_t, A_t)$ where each node $u \in V_t$ represents a joint $u$ in the skeleton and each edge $e \in E_t$ represents a bone connecting two joints, and $z_{u,t} \in A_t$ is a vector representing the attributes of joint $u$ including bone length, size, motor strength, etc. Here, the design $D_t$ is indexed by $t$ since it can be changed by transform actions during the transform stage.

**MDP with Design.** To accommodate the transform actions and changes in agent design $D_t$, we redefine the agent's MDP by modifying the state and action space as well as the transition dynamics. Specifically, the new state $s_t = (s_t^e, D_t, \Phi_t)$ includes the agent's state $s_t^e$ in the environment and the agent design $D_t = (V_t, E_t, A_t)$, as well as a stage flag $\Phi_t$. The new action $a_t \in \{a_t^d, a_t^e\}$ consists of both the transform action $a_t^d$ and the motor control action $a_t^e$. The new transition dynamics $\mathcal{T}(s_{t+1}^e, D_{t+1}, \Phi_{t+1} | s_t^e, D_t, \Phi_t, a_t)$ reflects the changes in the state and action.

**Transform2Act Policy.** The policy $\pi_\theta$ with parameters $\theta$ is a conditional policy that selects the type of actions based on which stage the agent is in:

$$\pi_\theta(a_t | s_t^e, D_t, \Phi_t) = \begin{cases} \pi_\theta^d(a_t^d | D_t, \Phi_t), & \text{if } \Phi_t = \text{Transform} \\ \pi_\theta^e(a_t^e | s_t^e, D_t, \Phi_t), & \text{if } \Phi_t = \text{Execution} \end{cases} \tag{7}$$

where two sub-policies $\pi_\theta^d$ and $\pi_\theta^e$ are used in the transform and execution stages respectively. As the agent in the transform stage does not interact with the environment, the transform sub-policy $\pi_\theta^d(a_t^d | D_t, \Phi_t)$ is not conditioned on the environment state $s_t^e$ and only outputs transform actions $a_t^d$. The execution sub-policy $\pi_\theta^e(a_t^e | s_t^e, D_t, \Phi_t)$ is conditioned on both $s_t^e$ and the design $D_t$ to output control actions $a_t^e$. $D_t$ is needed since the transformed design will affect the dynamics of the environment in the execution stage. As a notation convention, policies with different superscripts (e.g., $\pi_\theta^d$ and $\pi_\theta^e$) do not share the same set of parameters.

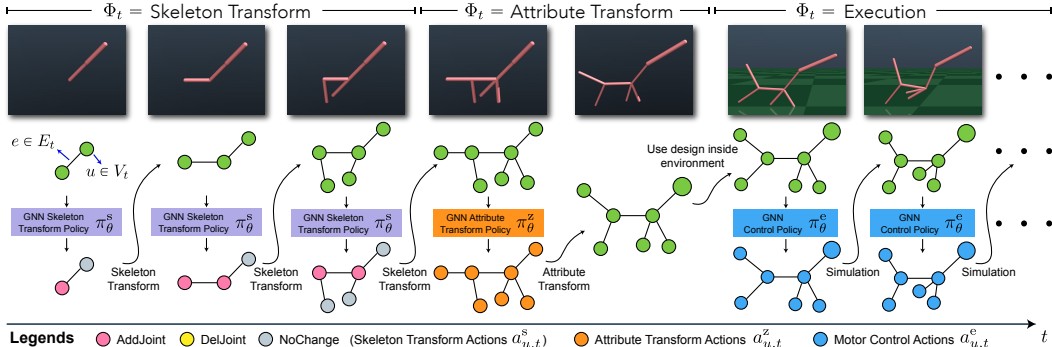

Figure 2: Transform2Act divides an episode into three stages: (1) Skeleton transform stage, where sub-policy $\pi_\theta^s$ changes the agent's skeleton by adding or removing joints; (2) Attribute transform stage, where sub-policy $\pi_\theta^z$ changes joint attributes (e.g., length, size); (3) Execution stage, sub-policy $\pi_\theta^e$ selects control actions for the newly-designed agent to interact with the environment.

## 4.1 TRANSFORM STAGE

In the transform stage, starting from an initial design $D_0$, the agent follows the transform sub-policy $\pi_\theta^d(a_t^d|D_t, \Phi_t)$ which outputs transform actions to modify the design. Since the design $D_t = (V_t, E_t, A_t)$ includes both the skeletal graph $(V_t, E_t)$ and joint attributes $A_t$, the transform action $a_t^d \in \{a_t^s, a_t^z\}$ consists of two types: (1) **Skeleton Transform Action** $a_t^s$, which is discrete and changes the skeletal graph $(V_t, E_t)$ by adding or deleting joints; (2) **Attribute Transform Action** $a_t^z$, which modifies the attributes of each joint and can be either continuous or discrete. Based on the two types of transform actions, we further divide the transform stage into two sub-stages – **Skeleton Transform Stage** and **Attribute Transform Stage** – where $a_t^s$ and $a_t^z$ are taken by the agent respectively. We can then write the transform sub-policy as conditioned on the agent's stage:

$$\pi_\theta^d(a_t^d|D_t, \Phi_t) = \begin{cases} \pi_\theta^s(a_t^s|D_t, \Phi_t), & \text{if } \Phi_t = \text{Skeleton Transform} \\ \pi_\theta^z(a_t^z|D_t, \Phi_t), & \text{if } \Phi_t = \text{Attribute Transform} \end{cases} \tag{8}$$

where the agent follows sub-policies $\pi_\theta^s$ for $N_s$ timesteps in the skeleton transform stage and then follows $\pi_\theta^z$ for $N_z$ timesteps in the attribute transform stage. It may be tempting to merge the two stages and apply skeleton and attribute transform actions together. However, we separate the two stages since the skeleton action can increase the number of joints in the graph, while attribute transform actions can only output attribute changes for existing joints.

**Skeleton Transform.** The skeleton transform sub-policy $\pi_\theta^s(a_t^s|D_t, \Phi_t)$ adopts a GNN-based network, where the action $a_t^s = \{a_{u,t}^s|u \in V_t\}$ is factored across joints and each joint $u$ outputs its categorical action distribution $\pi_\theta^s(a_{u,t}^s|D_t, \Phi_t)$. The policy distribution is the product of all joints' action distributions:

$$\pi_\theta^s(a_{u,t}^s|D_t, \Phi_t) = \mathcal{C}(a_{u,t}^s; l_{u,t}), \quad l_{u,t} = \text{GNN}_s(u, A_t; V_t, E_t), \quad \forall u \in V_t, \tag{9}$$

$$\pi_\theta^s(a_t^s|D_t, \Phi_t) = \prod_{u \in V_t} \pi_\theta^s(a_{u,t}^s|D_t, \Phi_t), \tag{10}$$

where the GNN uses the joint attributes $A_t$ as input node features to output the logits $l_{u,t}$ of each joint's categorical action distribution $\mathcal{C}$. The skeleton transform action $a_{u,t}^s$ has three choices:

- **AddJoint**: joint $u$ will add a child joint $v$ to the skeletal graph, which inherits its attribute $z_{u,t}$.
- **DelJoint**: joint $u$ will remove itself from the skeletal graph. The action is only performed when the joint $u$ has no child joints, which is to prevent aggressive design changes.
- **NoChange**: no changes will be made to joint $u$.

After the agent applies the skeleton transform action $a_{u,t}^s$ at each joint $u$, we obtain the design $D_{t+1} = (V_{t+1}, E_{t+1}, A_{t+1})$ for the next timestep with a new skeletal structure.

The GNN-based skeleton transform policy enables rapid growing of skeleton structures since every joint can add a child joint at each timestep. Additionally, it also encourages symmetric structures due to weight sharing in GNNs where mirroring joints can choose the same action.

---

**Algorithm 1:** Agent Design Optimization with Transform2Act Policy

---

initialize Transform2Act policy $\pi_\theta$ ;
**while** *not reaching max iterations* **do**
    memory $\mathcal{M} \leftarrow \emptyset$ ;
    **while** $\mathcal{M}$ *not reaching batch size* **do**
        $D_0 \leftarrow$ initial agent design ;
        **for** $t = 0, 1, \ldots, N_\mathrm{s} - 1$ **do**             `// Skeleton Transform Stage`
            sample skeleton transform action $a_t^\mathrm{s} \sim \pi_\theta^\mathrm{s}$; $\Phi_t \leftarrow$ Skeleton Transform ;
            $D_{t+1} \leftarrow$ apply $a_t^\mathrm{s}$ to modify skeleton $(V_t, E_t)$ in $D_t$ ;
            $r_t \leftarrow 0$; store $(r_t, a_t^\mathrm{s}, D_t, \Phi_t)$ into $\mathcal{M}$ ;
        **for** $t = N_\mathrm{s}, \ldots, N_\mathrm{s} + N_\mathrm{z} - 1$ **do**       `// Attribute Transform Stage`
            sample attribute transform action $a_t^\mathrm{z} \sim \pi_\theta^\mathrm{z}$; $\Phi_t \leftarrow$ Attribute Transform ;
            $D_{t+1} \leftarrow$ apply $a_t^\mathrm{z}$ to modify attributes $A_t$ in $D_t$ ;
            $r_t \leftarrow 0$; store $(r_t, a_t^\mathrm{z}, D_t, \Phi_t)$ into $\mathcal{M}$ ;
        $s_{N_\mathrm{s}+N_\mathrm{z}}^\mathrm{e} \leftarrow$ initial environment state ;
        **for** $t = N_\mathrm{s} + N_\mathrm{z}, \ldots, H$ **do**              `// Execution Stage`
            sample motor control action $a_t^\mathrm{e} \sim \pi_\theta^\mathrm{e}$; $\Phi_t \leftarrow$ Execution ;
            $s_{t+1}^\mathrm{e} \leftarrow$ environment dynamics $\mathcal{T}^\mathrm{e}(s_{t+1}^\mathrm{e}|s_t^\mathrm{e}, a_t^\mathrm{e})$; $D_{t+1} \leftarrow D_t$ ;
            $r_t \leftarrow$ environment reward; store $(r_t, a_t^\mathrm{s}, s_t^\mathrm{e}, D_t, \Phi_t)$ into $\mathcal{M}$ ;
    update $\pi_\theta$ with PPO using samples in $\mathcal{M}$             `// Policy Update`
**return** $\pi_\theta$

---

**Attribute Transform.** The attribute transform sub-policy $\pi_\theta^\mathrm{z}(a_t^\mathrm{z}|D_t, \Phi_t)$ adopts the same GNN-based network as the skeleton transform sub-policy $\pi_\theta^\mathrm{s}$. The main difference is that the output action distribution can be either continuous or discrete. In this paper, we only consider continuous attributes including bone length, size, and motor strength, but our method by design can generalize to discrete attributes such as geometry types. The policy distribution is defined as:

$$\pi_\theta^\mathrm{z}(a_{u,t}^\mathrm{z}|D_t, \Phi_t) = \mathcal{N}(a_{u,t}^\mathrm{z}; \mu_{u,t}^\mathrm{z}, \Sigma^\mathrm{z}), \quad \mu_{u,t}^\mathrm{z} = \mathrm{GNN}_\mathrm{z}(u, A_t; V_t, E_t), \quad \forall u \in V_t, \quad (11)$$

$$\pi_\theta^\mathrm{z}(a_t^\mathrm{z}|D_t, \Phi_t) = \prod_{u \in V_t} \pi_\theta^\mathrm{z}(a_{u,t}^\mathrm{z}|D_t, \Phi_t), \quad (12)$$

where the GNN outputs the mean $\mu_{u,t}^\mathrm{z}$ of joint $u$'s Gaussian action distribution and $\Sigma^\mathrm{z}$ is a learnable diagonal covariance matrix independent of $D_t, \Phi_t$ and shared by all joints. Each joint's action $a_{u,t}^\mathrm{z}$ is used to modify its attribute feature: $z_{u,t+1} = z_{u,t} + a_{u,t}^\mathrm{z}$, and the new design becomes $D_{t+1} = (V_t, E_t, A_{t+1})$ where the skeleton $(V_t, E_t)$ remains unchanged.

**Reward.** During the transform stage, the agent does not interact with the environment, because changing the agent's design such as adding or removing joints while interacting with the environment does not obey the laws of physics and may be exploited by the agent. Since there is no interaction, we do not assign any environment rewards to the agent. While it is possible to add rewards based on the current design to guide the transform actions, in this paper, we do not use any design-related rewards for fair comparison with the baselines. Thus, no reward is assigned to the agent in the transform stage, and the transform sub-policies are only trained using future rewards from the execution stage.

**Inference.** At test time, the most likely action will be chosen by both the skeleton and attribute transform policies. The design $D_t$ after the transform stage is the final design output.

## 4.2 EXECUTION STAGE

After the agent performs $N_\mathrm{s}$ skeleton transform and $N_\mathrm{z}$ attribute transform actions, it enters the execution stage where the agent assumes the transformed design and interacts with the environment. A GNN-based execution policy $\pi_\theta^\mathrm{e}(a_t^\mathrm{e}|s_t^\mathrm{e}, D_t, \Phi_t)$ is used in this stage to output motor control actions $a_t^\mathrm{e}$ for each joint. Since the agent now interacts with the environment, the policy $\pi_\theta^\mathrm{e}$ is conditioned on the environment state $s_t^\mathrm{e}$ as well as the transformed design $D_t$, which affects the dynamics of the environment. Without loss of generality, we assume the control actions are continuous. The

execution policy distribution is defined as:

$$\pi_\theta^e(a_{u,t}^e|s_t^e, D_t, \Phi_t) = \mathcal{N}(a_{u,t}^e; \mu_{u,t}^e, \Sigma^e), \quad \mu_{u,t}^e = \text{GNN}_e(u, s_t^e, A_t; V_t, E_t), \quad \forall u \in V_t, \quad (13)$$

$$\pi_\theta^e(a_t^e|s_t^e, D_t, \Phi_t) = \prod_{u \in V_t} \pi_\theta^e(a_{u,t}^e|s_t^e, D_t, \Phi_t), \quad (14)$$

where the environment state $s_t^e = \{s_{u,t}^e | u \in V_t\}$ includes the state of each node $u$ (e.g., joint angle and velocity). The GNN uses the environment state $s_t^e$ and joint attributes $A_t$ as input node features to output the mean $\mu_{u,t}^e$ of each joint's Gaussian action distribution. $\Sigma^e$ is a state-independent learnable diagonal covariance matrix shared by all joints. The agent applies the motor control actions $a_t^e$ to all joints and the environment transitions the agent to the next environment state $s_{t+1}^e$ according to the environment's transition dynamics $\mathcal{T}^e(s_{t+1}^e|s_t^e, a_t^e)$. The design $D_t$ remains unchanged throughout the execution stage.

### 4.3 VALUE ESTIMATION

As we use an actor-critic method (PPO) for policy optimization, we need to approximate the value function $\mathcal{V}$, i.e., the expected total discounted rewards starting from state $s_t = (s_t^e, D_t, \Phi_t)$:

$$\mathcal{V}(s_t^e, D_t, \Phi_t) \triangleq \mathbb{E}_{\pi_\theta}\left[\sum_{t=0}^{H} \gamma^t r_t\right]. \quad (15)$$

We learn a GNN-based value network $\hat{\mathcal{V}}_\phi$ with parameters $\phi$ to approximate the true value function:

$$\hat{\mathcal{V}}_\phi(s_t^e, D_t, \Phi_t) = \text{GNN}_v(\text{root}, s_t^e, A_t; V_t, E_t) \quad (16)$$

where the GNN takes the environment state $s_t^e$, joint attributes $A_t$ and stage flag $\Phi_t$ (one-hot vector) as input node features to output a scalar at each joint. We use the output of the root joint as the predicted value. The value network $\hat{\mathcal{V}}_\phi$ is used in all stages. In the transform stage, the environment state $s_t^e$ is unavailable so we set it to 0.

### 4.4 IMPROVE SPECIALIZATION WITH JOINT-SPECIALIZED MLP

As demonstrated in prior work (Wang et al., 2018; Huang et al., 2020), GNN-based control policies enjoy superb generalizability across different designs. The generalizability can be attributed to GNNs' weight sharing across joints, which allows new joints to share the knowledge learned by existing joints. However, weight sharing also means that joints in similar states will choose similar actions, which can seriously limit the transform policies and the per-joint specialization of the resulting design. Specifically, as both skeleton and attribute transform policies are GNN-based, due to weight sharing, joints in similar positions in the graph will choose similar or the same transform actions. While this does encourage the emergence of symmetric structures, it also limits the possibility of asymmetric designs such as different lengths, sizes, or strengths of the front and back legs.

To improve the per-joint specialization of GNN-based policies, we propose to add a joint-specialized multi-layer perceptron (JSMLP) after the GNNs. Concretely, the JSMLP uses different MLP weights for each joint, which allows the policy to output more specialized joint actions. To achieve this, we design a joint indexing scheme that is consistent across all designs and maintain a joint-indexed weight memory. The joint indexing is used to identify joint correspondence across designs where two joints with the same index are deemed the same. This allows the same joint in different designs to use the same MLP weights. However, we cannot simply index joints using a breadth-first search (BFS) for each design since some joints may appear or disappear across designs, and completely different joints can be assigned the same index in BFS. Instead, we index a joint based on the path from the root to the joint. The details of the indexing are provided in Appendix C. As we will show in the ablation studies, JSMLPs can improve the performance of both transform and control policies.

## 5 EXPERIMENTS

We design our experiments to answer the following questions: (1) Does our method, Transform2Act, outperform previous methods in terms of convergence speed and final performance? (2) Does Transform2Act create agents that look plausible? (3) How do critical components – GNNs and JSMLPs – affect the performance of Transform2Act?

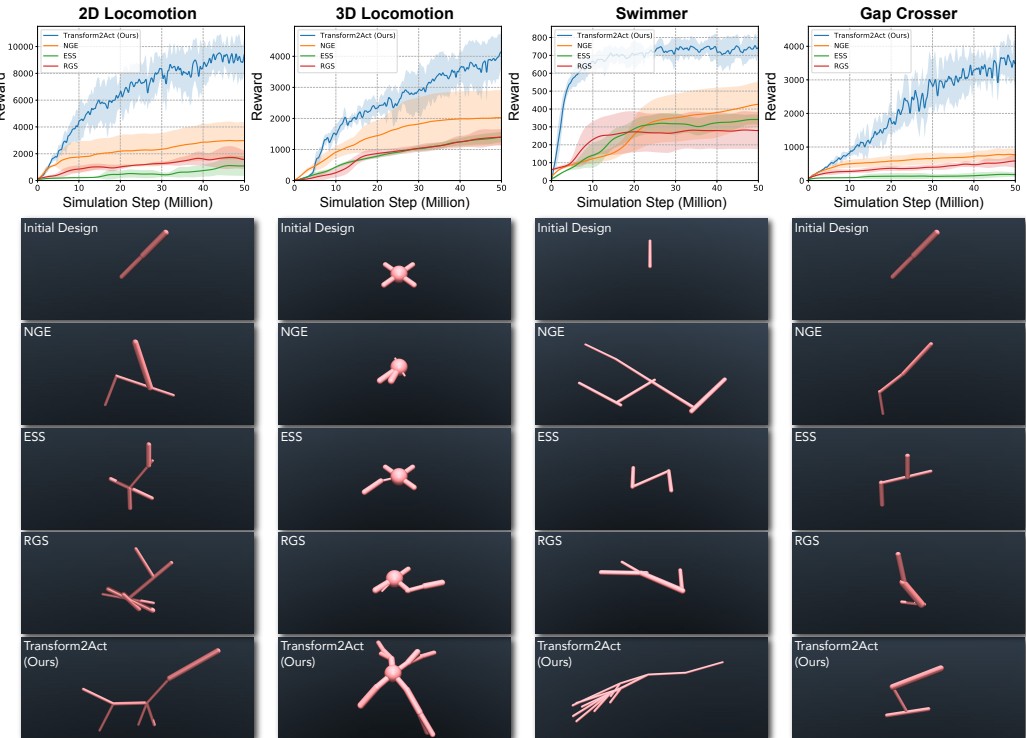

Figure 3: **Baseline comparison.** For each environment, we plot the mean and standard deviation of total rewards against the number of simulation steps for all methods, and show their final designs.

**Environments.** We evaluate Transform2Act on four distinct environments using the MuJoCo simulator (Todorov et al., 2012): (1) **2D Locomotion**, where a 2D agent living in an $xz$-plane is tasked with moving forward as fast as possible, and the reward is its forward speed. (2) **3D Locomotion**, where a 3D agent's goal is to move as fast as possible along $x$-axis and is rewarded by its forward speed along $x$-axis. (3) **Swimmer**, where a 2D agent living in water with 0.1 viscosity and confined in an $xy$-plane is rewarded by its moving speed along $x$-axis. (4) **Gap Crosser**, where a 2D agent living in an $xz$-plane needs to cross periodic gaps and is rewarded by its forward speed.

**Baselines.** We compare Transform2Act with the following baselines that also optimize both the skeletal structure and joint attributes of an agent: (1) **Neural Graph Evolution (NGE)** (Wang et al., 2019), which is an ES-based method that uses GNNs to enable weight sharing between an agent and its offspring. (2) **Evolutionary Structure Search (ESS)** (Sims, 1994), which is a classic ES-based method that has been used in recent works (Cheney et al., 2014; 2018). (3) **Random Graph Search (RGS)**, which is a baseline employed in (Wang et al., 2019) that trains a population of agents with randomly generated skeletal structures and joint attributes.

## 5.1 COMPARISON WITH BASELINES

In Figure 3 we show the learning curves of each method and their final agent designs for all four environments. For each method, the learning curve plots use six seeds per environment and plot the agent's total rewards against the total number of simulation steps used by the method. For ES-based baselines with a population of agents, we plot the performance of the best agent at each iteration. We can clearly see that our method, Transform2Act, consistently and significantly outperforms the baselines in terms of convergence speed and final performance.

Next, let us compare the final designs generated by each method in Figure 3. For better comparison, we encourage the reader to see these designs in video on the project website. For 2D locomotion, Transform2Act is able to discover a giraffe-like agent that can run extremely fast and remain stable. The design is suitable for optimizing the given reward function since it has a long neck to increase its forward momentum and balance itself when jumping forward. For 3D Locomotion, Transform2Act creates a spider-like agent with long legs. The design is roughly symmetric due to the GNN-based

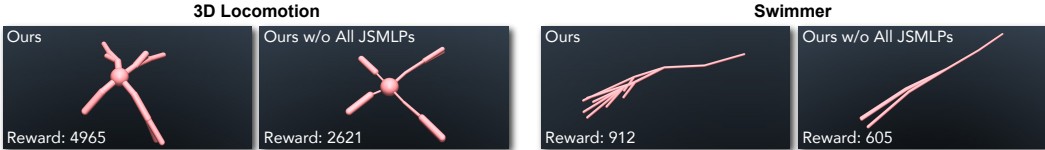

Figure 4: **Ablation studies.** The plots indicate that GNNs and JSMLPs both contribute to the performance and stability of our approach greatly.

Figure 5: **Effect of JSMLPs.** Designs without JSMLPs are overly symmetric with little per-joint specialization, which leads to worse performance.

transform policy, but it also contains joint-specific features thanks to the JSMLPs which help the agent attain better performance. For Swimmer, Transform2Act produces a squid-like agent with long bodies and numerous tentacles. As shown in the video, the movement of these tentacles propels the agent forward swiftly in the water. Finally, for Gap Crosser, Transform2Act designs a Hopper-like agent that can jump across gaps. Overall, we can see that Transform2Act is able to find plausible designs similar to giraffes, squids, and spiders, while the baselines fail to discover such designs and have much lower performance in all four environments. We also provide a detailed discussion of our method's advantages over the strong ES-based baseline, NGE (Wang et al., 2019), in Appendix B.

**Continuous Design Optimization.** In some cases, we may already have a good agent designed by experts and want to keep its skeletal structure. In such scenarios, we can also apply Transform2Act to finetune the expert design's attributes to further improve its performance. We show that our method outperforms a popular continuous design optimization approach (Ha, 2019) in Appendix A.

## 5.2 ABLATION STUDIES

We aim to investigate the importance of two critical components in our approach – GNNs and JSMLPs. We design three variants of our approach: (1) **Ours w/o GNNs**, where we remove all the GNNs from our Transform2Act policy and uses JSMLP only; (2) **Ours w/o Control JSMLP**, where we remove the JSMLP from our execution sub-policy $\pi_\theta^e$; (3) **Ours w/o All JSMLPs**, where we remove all the JSMLPs from the execution sub-policy $\pi_\theta^e$ and transform sub-policies $\pi_\theta^s$ and $\pi_\theta^z$. The learning curves of all variants are shown in Figure 4. It is evident that GNNs are a crucial part of our approach as indicated by the large decrease in performance of the corresponding variant. Additionally, JSMLPs are very instrumental for both design and control in our method, which not only increase the agent's performance but also make the learning more stable. For Swimmer, the variant without JSMLPs has a very large performance variance. To further study the effect of JSMLPs, we also show the design with and without JSMLPs for 3D Locomotion and Swimmer in Figure 5. We can observe that the designs without JSMLPs are overly symmetric and uniform while the designs with JSMLPs contain joint-specialized features that help the agent achieve better performance.

## 6 CONCLUSION

In this paper, we proposed a new transform-and-control paradigm that formulates design optimization as conditional policy learning with policy gradients. Compared to prior ES-based methods that randomly mutate designs, our approach is more sample-efficient as it leverages a parameterized policy to select designs based on past experience and also allows experience sharing across different designs. Experiments show that our approach outperforms prior methods in both convergence speed and final performance by an order of magnitude. Our approach can also automatically discover plausible designs similar to giraffes, squids, and spiders. For future work, we are interested in leveraging RL exploration methods to further improve the sample efficiency of our approach.

**Acknowledgements.** This work is supported by the NVIDIA Graduate Fellowship.

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

## A    COMPARISON WITH CONTINUOUS DESIGN OPTIMIZATION BASELINES

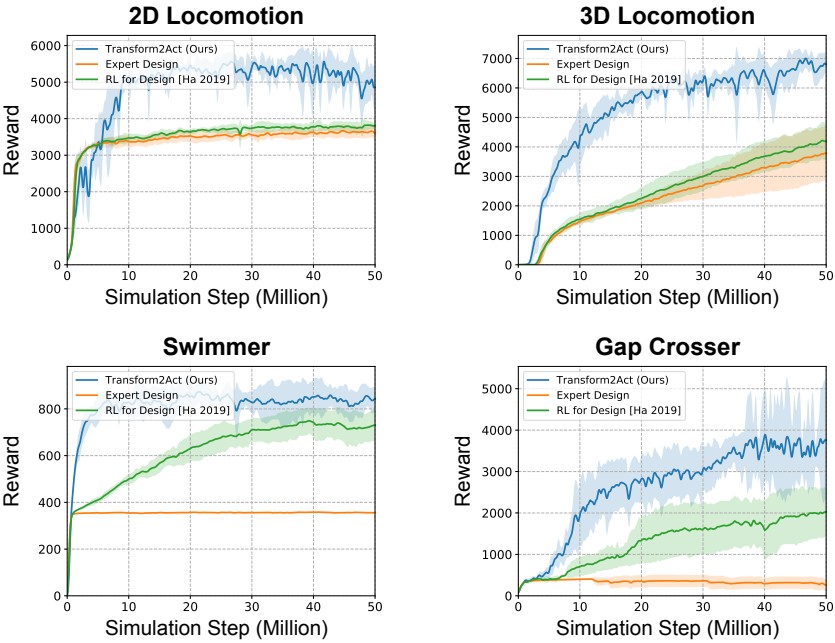

Figure 6: Baseline comparison of continuous design optimization for finetuning expert designs. The expert designs are taken from OpenAI Gym, where we use Hopper for 2D Locomotion and Gap Crosser, Ant for 3D Locomotion, and Gym Swimmer for our Swimmer.

In this section, we aim to evaluate our approach's ability to finetune a given expert design to further improve its performance. To this end, we perform experiments with the skeleton transform stage disabled in our approach and use the expert design as the initial design. The expert designs are taken from OpenAI Gym (Brockman et al., 2016), where we use Hopper for 2D Locomotion and Gap Crosser, Ant for 3D Locomotion, and Gym Swimmer for our Swimmer. We compare our method against a popular RL-based continuous design optimization approach (Ha, 2019). We use the same network architectures for all methods to ensure fair comparison. The policy for the expert design is trained with PPO (Schulman et al., 2017). The results are shown in Figure 6 where we use six seeds per environment for each method. We can see that our approach converges much faster than the expert design and baseline (Ha, 2019) for all environments. Our method also achieves significantly better final performance after 50 million steps of training. This demonstrates that, in addition to combinatorial design optimizatin, our approach also enjoys superior performance in continuous design optimization, which has many real-world applications.

## B    ADVANTAGES OF TRANSFORM2ACT OVER NGE

There are three main advantages of our method over NGE (Wang et al., 2019):

1. **NGE does not allow experience sharing among species in a generation.** As shown in Algorithm 1 of NGE, each species inside a generation $j$ has its own set of weights $\theta_i^j$ and is trained independently without sharing experiences among different species. The experience sharing in NGE is only enabled through weighting sharing between a species and its parent species from the previous generation. This means that if there are $N$ species in a generation, in every epoch, each species is only trained with $M/N$ number of experience samples where $M$ is the sample budget for each epoch. At the end of the training, each species has only used $EM/N$ samples for training where $E$ is the number of epochs. In contrast, in our method, every design shares the same control policy, so the policy can use all $EM$ samples. Therefore, our method allows better experience sharing across different designs, which improves sample efficiency.

2. **Our method uses a transform policy to change designs instead of random mutation.** Our transform policy takes the current design as input to output the transform actions (design changes). Through training, the policy learns to store information about which design to prefer and which design to avoid. This information is also shared among different joints via the use of GNNs, where joints in similar states choose similar transform actions (which is also balanced by JSMLPs for joint specialization). Additionally, the policy also allows every joint to simultaneously change its design (e.g., add a child joint, change joint attributes). For example, in 3D Locomotion, the agent can simultaneously grow its four feet in a single timestep, while ES-based methods such as NGE will take four different mutations to obtain four feet. Therefore, our method with the transform policy allows better generalization and experience sharing among joints, compared to ES-based methods that perform random mutation.

3. **Our method allows more exploration.** Our transform-and-control policy tries out a new design every episode, which means our policy can try $M/H_{\text{avg}}$ designs every epoch, where $M$ is the total number of sample timesteps and $H_{\text{avg}}$ is the average episode length. There is also more exploration for our approach at the start of the training when $H_{\text{avg}}$ is small. On the other hand, ES-based methods such as NGE only try $N$ (num of species) different designs every epoch. If NGE uses too many species (large $N$), each species will have few samples to train as mentioned in point 1. Therefore, $N$ is typically set to be $\ll M/H_{\text{avg}}$. For example, $N$ is set from 16 to 100 in NGE, while $M/H_{\text{avg}}$ in our method can be more than 2000.

## C DETAILS OF JOINT-SPECIALIZED MLP (JSMLP)

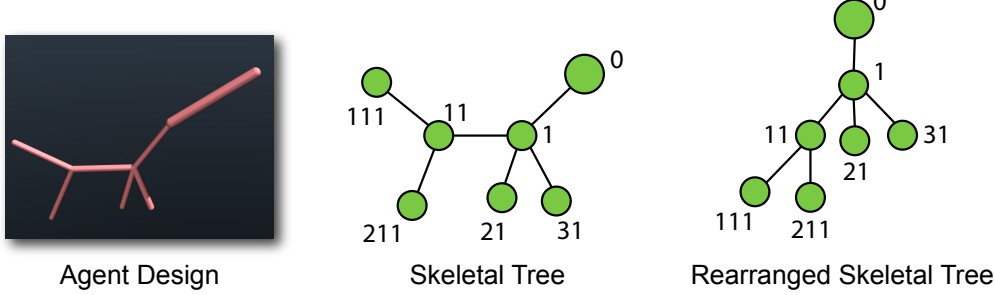

| Agent Design | Skeletal Tree | Rearranged Skeletal Tree |

Figure 7: An example of the joint indexing used by the JSMLPs in our approach. Every non-root joint is indexed by the path from the root to the joint. Starting from an empty index string at the root, every time we go to the $i$-th child of the current joint, we add $i$ to the left side of the index string. For example, the joint '211' gets its index because to reach the joint from the root, we need to first go the first child '1' of the root, and then go to the first child '11' of joint '1', and then go to the second child '211' of joint '11', which is the target joint.

As discussed in Section 4.4, JSMLPs use different sets of MLP weights for each joint to improve the per-joint specialization ability of our GNN-based policy networks. To achieve this, we need a joint indexing to retrieve MLP weights from the joint-indexed weights memory. The joint indexing is used to identify joint correspondence across designs where two joints with the same index are deemed the same. This allows the same joint in different designs to use the same MLP weights.

Figure 7 illustrates our joint indexing method. We first assign '0' to the root joint. For any non-root joint, its index is based on the path from the root to the joint. Starting from an empty string at the root, we follow the path to the target joint (to be indexed) down the skeletal tree, and every time we go to the $i$-th child of the current joint, we add $i$ to the left side of the index string. For example, to reach the rightmost joint from the root, we first need to go to the first child of the root, which adds '1' to the empty string and obtains '1'; we then need to go to the third child of the current joint '1', which adds '3' to the current string '1' and obtains '31', which is the final index string for the rightmost joint since we have reached it. When a joint is removed and the design is changed, we reindex the joints based on the updated skeletal graph. We can also very easily convert the index string into an integer using base $N_C + 1$ where $N_C$ is the maximum number of children each joint is allowed to have.

# D  VISUALIZATIONS OF TRANSFORM ACTIONS

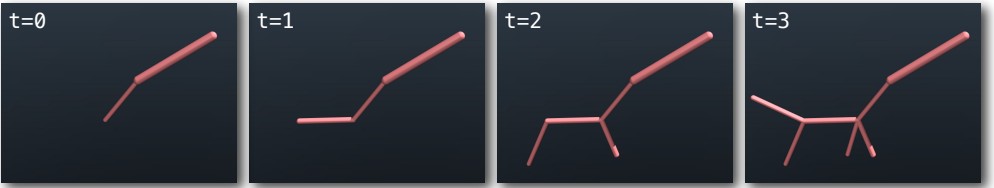

Figure 8: Designs process of Transform2Act for the 2D Locomotion environment. We visualize the design after each skeleton transform action. For better visualization, we use the joint attributes after the attribute transform stage.

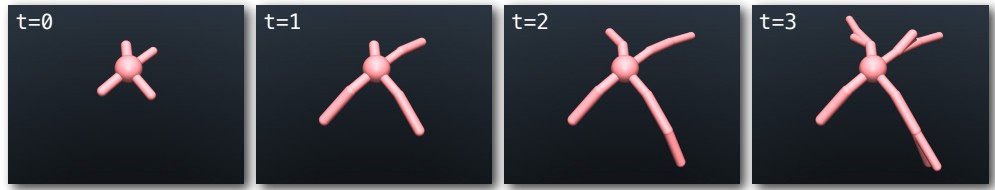

Figure 9: Designs process of Transform2Act for the 3D Locomotion environment. We visualize the design after each skeleton transform action. For better visualization, we use the joint attributes after the attribute transform stage.

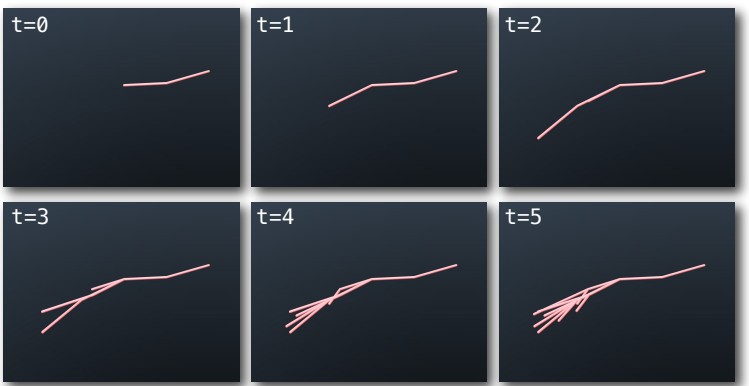

Figure 10: Designs process of Transform2Act for the Swimmer environment. We visualize the design after each skeleton transform action. For better visualization, we use the joint attributes after the attribute transform stage.

# E  ENVIRONMENT DETAILS

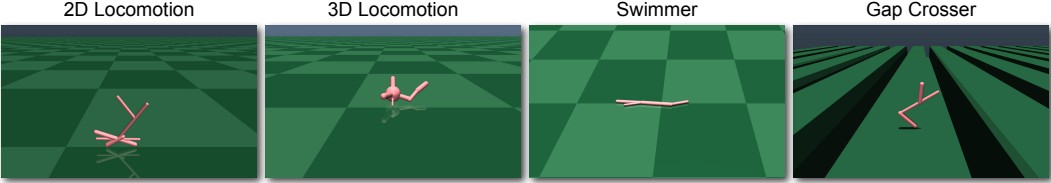

Figure 11: A random agent in each of the four environments.

In this section, we provide additional details about the four environments we use in our experiments. We show what a random agent looks like in each environment in Figure 11. Each joint of the agent uses a hinge connection with its parent joint. Each joint's environment state $s_{u,t}^{\text{e}}$ includes its joint angle and velocity. For the root joint, we also include other information such as the height, phase and world velocity. Zero padding is used to ensure that each joint's state has the same length. Each joint's attribute feature $z_{u,t}$ includes the bone vector, bone size, and motor gear value of the joint. Each attribute is normalized within the range $[-1, 1]$ using a loosely defined attribute range.

**2D Locomotion.** The agent in this environment lives inside an $xz$-plane with a flat ground at $z = 0$. Each joint of the agent is allowed to have a maximum of three child joints. For the root joint, we add its height and 2D world velocity to the environment state. The reward function is defined as

$$r_t = \left| p_{t+1}^x - p_t^x \right| / \delta t + 1 \,, \tag{17}$$

where $p_t^x$ denotes the $x$-position of the agent and $\delta t = 0.008$ is the time step. An alive bonus of 1 is also used inside the reward. The episode is terminated when the root height is below 0.7.

**3D Locomotion.** The agent in this environment lives freely in a 3D world with a flat ground at $z = 0$. Each joint of the agent is allowed to have a maximum of two child joints except for the root. For the root joint, we add its height and 3D world velocity to the environment state. The reward function is defined as

$$r_t = \left| p_{t+1}^x - p_t^x \right| / \delta t - w \cdot \frac{1}{J} \sum_{u \in V_t} \| a_{u,t}^{\text{e}} \|^2 \,,, \tag{18}$$

where $w = 0.0001$ is a weighting factor for the control penalty term, $J$ is the total number of joints, and the time step $\delta t$ is 0.04.

**Swimmer.** The agent in this environment lives in water with 0.1 viscosity and is confined within an $xy$-plane contacting a flat ground. Each joint of the agent is allowed to have a maximum of three child joints. For the root joint, we add its 2D world velocity to the environment state. The reward function is the same as the one used by 3D Locomotion as defined in Equation (18).

**Gap Crosser.** The agent in this environment lives inside an $xz$-plane. Unlike other environments, the terrain of this environment includes a periodic gap of 0.96 width with the period being 3.2. The height of the terrain is at 0.5. The agent must cross these gaps to move forward. Each joint of the agent is allowed to have a maximum of three child joints. For the root joint, we add its height, 2D world velocity, and a phase variable encoding the periodic $x$-position of the agent to the environment state. The reward function is defined as

$$r_t = \left| p_{t+1}^x - p_t^x \right| / \delta t + 0.1 \,, \tag{19}$$

where the time step $\delta t$ is = 0.008. An alive bonus of 0.1 is also used inside the reward. The episode is terminated when the root height is below 1.0.

**Other Details.** Since MuJoCo agents are specified using XML strings, during the transform stage, we represent each design as an XML string and modify its content based on the transform actions. At the start of the execution stage, the modified XML string is used to reset the MuJoCo simulator and load the newly-designed agent.

## F    COMPUTATION COST

Following standard RL practices, we use distributed trajectory sampling with mulitple CPU threads to accelerate training. For all the environments used in the paper, it takes around one day to train our model on a standard server with 20 CPU cores and an NVIDIA RTX 2080 Ti GPU.

## G    HYPERPARAMETERS AND TRAINING PROCEDURES

In this section, we present the hyperparameters searched and used for our method in Table 1 and the hyperparameters for the baselines in Table 2. We use PyTorch (Paszke et al., 2019) to implement all the models. For the implementation of GNNs, we employ the PyTorch Geometric package (Fey & Lenssen, 2019) and use GraphConv (Morris et al., 2019) as the GNN layer. When training the policy with PPO, we also adopt generalized advantage estimation (GAE) (Schulman et al., 2015).

For the baselines, we implement our own versions of NGE, RGS, and ESS following the publicly released code[1] of NGE (Wang et al., 2019). For NGE, we use the same GNN architecture as the one used in our method to make fair comparison. We also ensure that the baselines and our method use the same number of simulation steps for optimization. For instance, our method optimizes the policy with batch size 50000 for 1000 epochs, which amounts to 50 million simulation steps. The baselines use a population of 20 agents, and each is trained with a batch size of 20000 for 125 generations, which also amounts to 50 million simulation steps.

| Hyperparameter | Values Searched & **Selected** |
| --- | :---: |
| Num. of Skeleton Transforms $N_{\mathrm{s}}$ | 3, **5**, 10 |
| Num. of Attribute Transforms $N_{\mathrm{z}}$ | **1**, 3, 5 |
| Policy GNN Layer Type | **GraphConv** |
| JSMLP Activation Function | **Tanh** |
| GNN Size (Skeleton Transform) | (32, 32, 32), **(64, 64, 64)**, (128, 128, 128), (256, 256, 256) |
| JSMLP Size (Skeleton Transform) | **(128, 128)**, (256, 256), (512, 256), (256, 256, 256) |
| GNN Size (Attribute Transform) | (32, 32, 32), **(64, 64, 64)**, (128, 128, 128), (256, 256, 256) |
| JSMLP Size (Attribute Transform) | **(128, 128)**, (256, 256), (512, 256), (256, 256, 256) |
| GNN Size (Execution) | (32, 32, 32), **(64, 64, 64)**, (128, 128, 128), (256, 256, 256) |
| JSMLP Size (Execution) | **(128, 128)**, (256, 256), (512, 256), (256, 256, 256) |
| Diagonal Values of $\Sigma^{\mathrm{z}}$ | 1.0, 0.04, **0.01** |
| Diagonal Values of $\Sigma^{\mathrm{e}}$ | **1.0**, 0.04, 0.01 |
| Policy Learning Rate | **5e-5**, 1e-4, 3e-4 |
| Value GNN Layer Type | **GraphConv** |
| Value Activation Function | **Tanh** |
| Value GNN Size | (32, 32, 32), **(64, 64, 64)**, (128, 128, 128), (256, 256, 256) |
| Value MLP Size | (128, 128), (256, 256), **(512, 256)**, (256, 256, 256) |
| Value Learning Rate | 1e-4, **3e-4** |
| PPO clip $\epsilon$ | **0.2** |
| PPO Batch Size | 10000, 20000, **50000** |
| PPO Minibatch Size | 512, **2048** |
| Num. of PPO Iterations Per Batch | 1, 5, **10** |
| Num. of Training Epochs | **1000** |
| Discount factor $\gamma$ | 0.99, **0.995**, 0.997, 0.999 |
| GAE $\lambda$ | **0.95** |

Table 1: Hyperparameters searched and used by our method. The bold numbers among multiple values are the final selected ones. For Gap Crosser, we use 0.999 for the discount factor $\gamma$.

---

[1] https://github.com/WilsonWangTHU/neural_graph_evolution

| Hyperparameter | Values Searched & **Selected** |
|---|:---:|
| Num. of Generations | **125** |
| Agent Population Size | 10, **20**, 50, 100 |
| Elimination Rate | **0.15**, 0.2, 0.3, 0.4 |
| GNN Layer Type | **GraphConv** |
| MLP Activation | **Tanh** |
| Policy GNN Size | (32, 32, 32), **(64, 64, 64)**, (128, 128, 128) |
| Policy MLP Size | **(128, 128)**, (256, 256), (512, 256) |
| Policy Log Standard Deviation | **0.0**, -1.6, -2.3 |
| Policy Learning Rate | **5e-5**, 1e-4, 3e-4 |
| Value GNN Size | (32, 32, 32), **(64, 64, 64)**, (128, 128, 128) |
| Value MLP Size | (128, 128), (256, 256), **(512, 256)** |
| Value Learning Rate | 1e-4, **3e-4** |
| PPO clip $\epsilon$ | **0.2** |
| PPO Batch Size | 10000, **20000**, 50000 |
| PPO Minibatch Size | 512, **2048** |
| Num. of PPO Iterations Per Batch | 1, 5, **10** |
| Discount factor $\gamma$ | 0.99, **0.995**, 0.999 |
| GAE $\lambda$ | **0.95** |

Table 2: Hyperparameters searched and used by the baselines. The bold numbers among multiple values are the final selected ones. For Gap Crosser, we use 0.999 for the discount factor $\gamma$.

