# OpenReview forum: "Transform2Act: Learning a Transform-and-Control Policy for Efficient Agent Design"
_ICLR.cc/2022/Conference — ICLR 2022 Oral_

### Official Review · Reviewer_42PW · 2021-10-27

**Correctness:** 3
**Technical Novelty And Significance:** 4
**Empirical Novelty And Significance:** 3
**Recommendation:** 8
**Confidence:** 4

**Main Review:**

**POSITIVES**

To the reviewers knowledge (which is admittedly limited), this is the first attempt to learn the morphology of a robot this straight forward with RL. The method is easy to understand and improvements over evolutionary baselines appear very significant (even for 3 seeds).

**NEGATIVES**

The reviewer has 4 major complaints:

(1) The reward of the tested environments is not scale invariant. When the agent is allowed to construct a robot with twice the dimensions and torque, the robot will move (approximately) twice as fast and therefore get twice the reward. The reviewer assumes that there is a build in penalty for using large torques, but this is a delicate balancing act that the authors need to at least discuss. An additional figure that shows how the robots' size changes during training would reveal a systematic growth of the robot and put this reviewers mind at ease if that is not the case.

(2) The results are based on 3 random seeds. While this is (sadly) all to common in RL, it severely limits the conclusions one can draw with any certainty from the results. The performance gap to the baselines appears large enough to be significant, and the same holds for the "ours w/o GNNs" ablation, but the ablations without JSMLP have barely non-overlapping standard deviations, which is an insufficient measure of statistical significance for 3 seeds anyway. To make the presented claims about the ablations, the authors need at least (!) twice the number of seeds (if not more).

(3) The JSMLP method is complicated and poorly explained. It is also not clear to the reviewer why asymmetry is inherently favorable for robotics design. Even if one would like to include the possibility, the presented JSMLP method seems very ad-hoc and might have some unintended side effects. For example, the decision to remove one node can change the node index (e.g. from 31 to 21 in Figure 6) without changing anything about the node. The authors are encouraged to look into Relational GCN as an alternative to represent asymmetric morphologies.

(4) It is unclear whether GNN actually generalize well (in particular out-of-distribution) over different robot morphologies. Kurin et al. (2021) have shown some very convincing counter-examples and suggest to use attention instead of GNN layers. The authors must discuss this alternative and are encouraged to evaluate their experiments with attention layers to see whether this yields any improvement.

**DETAILED COMMENTS**

- $r_t$ must be defined
- clarify that in the definition of J, H is a variable (due to episodic environment)
- when you introduce the design D, doesn't it also affect the state and action space?
- you do not explicitly state in the main paper which variant of GNN you use, e.g. which aggregation function
- (eq.7) looks as if $a^e$ and $a^d$ can be executed simultaneously instead of alternatively. Better select one $a$ (over the union of the action spaces).
- it is not clear why you allow the removal of joints, as it could lead to cycles
- it would be good to have a "stop changing the morphology" action instead of always running for $N_s$ and $N_z$ actions
- the "Policy Update" line in Algorithm 1 must be in the first while loop, as PPO first collects the data set M and then uses it to update the model (currently it is called after every episode)
- you should mention before page 7 that your graph has a root node
- it would be nice to run in Figure 3 PPO on some standard morphology in each environment, for example HalfCheetah, Ant, Swimmer and Hopper, to see that your algorithm is a "super-human" designer. In particular Ant would make for an interesting comparison

**REFERENCES**

Kurin, V.; Igl, M.; Rocktaeschel, T.; Boehmer, W., and Whiteson, S. My body is a cage: the role of morphology in graph-based incompatible control. In International Conference on Learning Representations (ICLR), 2021. URL https://openreview.net/forum?id=N3zUDGN5lO.

**Summary Of The Paper:**

The paper poses the problem of morphology design for robots as RL training of one joint GNN policy. Their policy first generates the robot's morphology and then evaluates the design with a common behavior policy that conditions on it. The authors also introduce a technique called JSMPL to allow asymmetric morphologies. Experiments in the Mujoco simulator demonstrate a large improvement over evolutionary methods in terms of sample efficiency and final performance (although the latter is less clear, as neither method has clearly converged). Ablation studies show that the GNN architecture is essential, but are less clear about the impact of JSMPL.


**Summary Of The Review:**

The paper introduces the (to the reviewers knowledge) novel concept of learning the morphology with RL instead of evolution. The experiments need more seeds, but the advantage of the presented method appears significant. The method to allow asymmetric morphologies appears less significant and one might find a more elegant way to ensure this property (if it is at all needed).

The reviewer would be willing to increase the score if the authors could substantiate point (1) and promise to discuss (or convincingly argue against) the other points of criticism.

**POST-REBUTTAL**

While the response does not answer 100% of this reviewers questions, the resulting paper is nonetheless interesting enough, and the results clear enough, to merit publication.  The reviewer is therefore raising the score to 8.

---

> ### Author Response · Authors · 2021-11-22
> **Response to Reviewer 42PW [1/2]**
>
> Thank you for the constructive feedback, suggestions, and pointers. We also appreciate the acknowledgment of our approach's novelty. We have revised the paper to include experiments with more seeds and baseline comparison of continuous design optimization for finetuning expert designs (please see the revision summary post for all changes). Here, we aim to address the questions raised in the review.
>
> ---
>
> > **Q1:** *"The reward of the tested environments is not scale invariant..."*
>
> For all the environments, there is an upper bound on the bone length, which is close to the initial bone length. For example, for 2D Locomotion and Gap Crosser, the upper bound is 0.6 while the initial bone length is 0.4. Therefore, there is not much room for the agent to exploit the bone length. We also have a torque penalty built in for 3D Locomotion and Swimmer. We’d also like to note that all the methods in the paper use the same reward function and design space for fair comparison.
>
> ---
>
> > **Q2:** *"The results are based on 3 random seeds..."*
>
> We follow the suggestion and run additional experiments to report results with a total of 6 seeds in the revised version. We hope the reviewer can understand that these new experiments cost a significant amount of compute, i.e., 100+ machine days in total, which limits the number of experiments we can run.
>
> ---
>
> > **Q3:** *"The JSMLP method is complicated and poorly explained. It is also not clear to the reviewer why asymmetry is inherently favorable for robotics design. Even if one would like to include the possibility, the presented JSMLP method seems very ad-hoc and might have some unintended side effects..."*
>
> - We’d like to first motivate the need for asymmetry using the giraffe-like design obtained by our method for 2D Locomotion. If we look at the 3 grandchildren of the root joint, i.e., the two front legs and the torso, they are asymmetric. Without allowing any asymmetry in the design, it would be **impossible** to grow these different joints (torso and front legs) from the root since they are topologically equivariant, where the transform policy will treat these joints the same and output the same design changes. Similarly, if we want to design a humanoid from a chest joint, we would also need to grow asymmetric joints to develop various branches of body parts (arms, neck, and lower body).
> - For the JSMLPs, we'd like to politely argue that its core idea is very simple and not ad-hoc, i.e., combining weight-sharing layers (GNNs, or attention layers as suggested) with non-weight-sharing layers (MLPs) together to balance the generalization and specialization across joints for the policies. With more (6) random seeds, we also empirically show that this solution is both simple and effective in boosting the performance and learning stability of our method. We also believe the problem of generalization vs. specification raised in JSMLPs in these node weight sharing settings would be worth considering for future work.
> - For the special case mentioned by the reviewer, where removing sibling joints could change a joint’s index, we can easily prevent that case by only allowing to remove the sibling joint with the largest index so that it won’t affect other joints.
>
> ---
>
> > **Q4:** *"It is unclear whether GNN actually generalize well (in particular out-of-distribution) over different robot morphologies. Kurin et al. (2021) have shown some very convincing counter-examples and suggest to use attention instead of GNN layers..."*
>
> Thank you for pointing out this alternative, which we believe is a promising future direction. We have discussed this option in the related work of the revised version. We’d like to mention that our approach by design is not limited to GNNs and can use other node weight sharing mechanisms such as the suggested attention layers from [1]. Due to limited amount of computational resources, we have prioritized other experiments (e.g., more random seeds and continuous design optimization), but we will try attention layers too to see if they can yield better performance than GNNs.
>
> ---
>
> ### Answers to detailed comments
> Thank you for all the detailed comments which are very useful! Many comments are directly addressed in the revised paper. Here we provide answers to those that need explanations.
>
> ---
>
> > **Q5:** *"when you introduce the design D, doesn't it also affect the state and action space?"*
>
> The design $D$ will indeed affect the state and action space, and we have redefined the MDP in the "MDP with Design" paragraph in the initial submission.
>
> ---
>
> > **Q6:** *"it is not clear why you allow the removal of joints, as it could lead to cycles"*
>
> Allowing the removal of joints is mainly for use cases where we want to finetune a given initial design.
>
> ---
>
> > **Q7:** *"it would be good to have a 'stop changing the morphology' action"*
>
> This is a brilliant idea and would be interesting for future work to further improve efficiency.

---

> > ### Author Response · Authors · 2021-11-22
> > **Response to Reviewer 42PW [2/2]**
> >
> > > **Q8:** *"...PPO on some standard morphology in each environment..."*
> >
> > In Appendix A of the revised version, we perform experiments of training expert designs (provided by Gym) with PPO. By examining both Fig. 3 and 6, we can see that the designs obtained by our approach from scratch do outperform the expert designs in all environments.
> >
> > ---
> >
> > ### **Reference**
> >
> > [1] Kurin et al., "My Body Is A Cage: The Role of Morphology in Graph-based Incompatible Control." ICLR 2021.

---

> > > ### Comment · Reviewer_42PW · 2021-11-29
> > > **Good enough**
> > >
> > > Thanks to the authors for their response and changes.
> > > - The response to Q1 is not 100% convincing, as there are ways to overcome the limits of the original (expert-) design by increasing all bones' length and adding more joints to increase the limbs. However, the reviewer acknowledges that the comparison between morphology construction methods is still fair and that the resulting morphologies do not seem to exploit this (which raises the question why not).
> > > - While the authors very convincingly argued for asymmetry in Q3, it is still unclear why their JSMLP method provides a robust framework to induce this (e.g. how it generalizes when when nodes are deleted). This raises serious questions on how general this method is, e.g. when applied to another GNN task. However, it is worth mentioning that the authors are not alone in this (see https://openreview.net/forum?id=fy_XRVHqly for another approach that remains similarly unclear).
> > > - Some comments seem to have been misunderstood: the background section must clarify that $r_t = R(s_t, a_t)$ and that $D \in \mathcal D$ does not only change the transition model (as in the current version), but also the state and the action space.
> > >
> > > While the response does not answer 100% of this reviewers questions, the resulting paper is nonetheless interesting enough, and the results clear enough, to merit publication.  The reviewer is therefore raising the score to 8.

---

### Official Review · Reviewer_ZAQQ · 2021-11-01

**Correctness:** 4
**Technical Novelty And Significance:** 3
**Empirical Novelty And Significance:** 3
**Recommendation:** 8
**Confidence:** 4

**Main Review:**

Strengths: clear idea, described well, convincing experiments

Weaknesses:
1) Formulations of some claims could be made more neutral. E.g., "improves sample efficiency tremendously" -> substantially, considerably, significantly
2) The paper refers to policy gradients (PG) as "first-order" optimization methods. This may not be exactly technically correct as these methods still only use zero order information about the objective function. The authors are encouraged to consult the recent literature on the analysis of PG methods, e.g., [1].
3) Table 1 in Appendix E shows that N_s=5 skeleton transforms and N_z=1 attribute transform were selected using hyperparameter search. What qualitative results does one obtain if there are more than 1 attribute transform? Why allowing more skeleton transforms performs worse? In principle, it should provide more flexibility. Adding a discussion explaining the effects of these numbers to the main body of the paper would be illuminating.
4) Table 1 in Appendix E shows values -2.3 and 0.0 are selected for diagonal values of covariance matrices Sigma^z and Sigma^e. This seems to be a mistake, because diagonal values should be greater than zero.

[1] Agarwal, A., Henaff, M., Kakade, S., & Sun, W. (2020). Pc-pg: Policy cover directed exploration for provable policy gradient learning. arXiv preprint arXiv:2007.08459.



**Summary Of The Paper:**

The paper proposes an algorithm for simultaneous agent design and policy optimization. The choice of the body structure is treated as another action available to the agent. Therefore, the policy is parameterized by graph neural networks (GNNs), and it outputs i) the skeleton structure, ii) node attributes such as bone length, size, motor strength, and iii) motor control commands. Thanks to the parameterization via GNNs, the policy can be trained with PPO. Experiments show that the proposed method outperforms prior approaches, which mainly employ evolutionary methods for optimization, whereas the proposed method leverages more sample efficient policy gradient algorithms.

**Summary Of The Review:**

Strong paper. The proposed approach is novel and can be impactful in other discrete-continuous optimization domains. Experiments validate the advantages of the proposed method. The paper is clear and easy to follow.

====

I am satisfied with the response of the authors and I maintain my score "8: accept, good paper".

---

> ### Author Response · Authors · 2021-11-22
> **Response to Reviewer ZAQQ**
>
> Thank you for the constructive feedback and appreciation of our approach's novelty and potential impact. We have revised the paper to include experiments with more seeds and baseline comparison of continuous design optimization for finetuning expert designs (please see the revision summary post for all changes). Here, we aim to address the questions raised in the review.
>
> ---
>
> > **Q1:** *"Formulations of some claims could be made more neutral..."*
>
> Thank you for the suggestion, we have modified the claims accordingly.
>
> ---
>
> > **Q2:** *"The paper refers to policy gradients (PG) as 'first-order' optimization methods. This may not be exactly technically correct as these methods still only use zero order information about the objective function."*
>
> We acknowledge that this statement is a bit confusing, and we have modified it in the revised version. What we intended to say by “first-order” is that our method uses a parametrized transform policy to output design-changing actions, which enables us to use policy gradients methods for policy optimization, and policy gradients use the first-order information in the policy by taking the exact gradient of the policy w.r.t. its parameters. Our method, using a parametrized transform policy to output design changes, is more efficient because the policy allows better generalization and experience sharing among joints, where joints in similar states choose similar transform actions using the policy. Please see Appendix B of the revised version for a detailed discussion of our approach vs. ES-based methods.
>
> ---
>
> > **Q3:** *"Q Table 1 in Appendix E shows that N_s=5 skeleton transforms and N_z=1 attribute transform were selected using hyperparameter search. What qualitative results does one obtain if there are more than 1 attribute transform? Why allowing more skeleton transforms performs worse? In principle, it should provide more flexibility. Adding a discussion explaining the effects of these numbers to the main body of the paper would be illuminating."*
>
> The reason for $N_\mathrm{z}=1$ attribute transform is that one step of attribute transform (adding a residual attribute) can already explore the whole space of the continuous design parameters. In our experiments, we found that using more attribute transforms ($N_\mathrm{z}>1$) yields quite similar performance. Therefore, we use $N_\mathrm{z}=1$ for efficiency.
>
> ---
>
> > **Q4:** *"Table 1 in Appendix E shows values -2.3 and 0.0 are selected for diagonal values of covariance matrices..."*
>
> These are indeed typos and thank you for spotting them. -2.3 and 0.0 are actually the log standard deviation, so the diagonal values of the covariance matrices should be 0.01 and 1.0 respectively.

---

### Official Review · Reviewer_mvHK · 2021-11-02

**Correctness:** 4
**Technical Novelty And Significance:** 3
**Empirical Novelty And Significance:** 3
**Recommendation:** 8
**Confidence:** 5

**Main Review:**

Strengths:
- Formulating design and control co-optimization as one sequential decision-making problem is novel.
- The paper's ideas all make sense (transform-and-control policy, skeleton and attribute transform, JSMLP).
- The empirical results look stronger than existing baselines.
- The paper is written to be easy to understand.

Weaknesses:
- Experiments are only conducted on four custom environments. Why not use existing environments from NGE or [1] (see references below)? Also, three random seeds are far below the standard.
- Little analysis on the empirical results, given no theoretical justification of the algorithm. The analysis can be further enhanced from several aspects: 1) Discussing comparison with NGE (probably the strongest baseline) about similarities and differences and how those differences lead to a huge performance increase; 2) Enriching ablation studies by training with skeleton transformation disabled and attribute transformation disabled respectively.

Concerns:
- I can't entirely agree with the argument that this approach enables first-order optimization of agent design. Technically, both this approach and ES-based methods do not have access to the ground-truth gradient and estimate first-order gradients based on the gathered experiences. So, this approach is still zeroth-order optimization, and it's not appropriate to claim that the sample efficiency comes from the first-order nature of the method.
- While I agree that ES-based methods have a high-dimensional search space for design, your approach does not essentially reduce that search space. Instead, the search space for policy is much larger in your formulation, i.e., the dimension of MDP becomes much higher.
- Could you provide more intuition on the comparison against NGE? By looking at the performance curves, it outperforms NGE by a large margin even without all JSMLPs. Is it because of the new formulation of the design optimization, or the attribute transform is optimized (unlike sampling from uniform distributions in NGE), or other reasons?
- I'm skeptical about the reported performance of NGE because of several reasons: 1) In the original NGE's paper, it outperforms RGS by a large margin, while in Figure 3 of this paper, the improvement is marginal; 2) I believe NGE should be much better than RGS is because NGE also uses GNN policies that allow experience sharing across different designs; 3) If experience sharing via GNN is not effective in NGE, why this is effective in your approach as you claimed? A good way to address my concerns is to probably run NGE's original implementation besides your own implementation and report the performance.

Other suggestions:
- Section 4 can be shortened to include more analysis in Section 5.
- In many practical use cases, probably we already have a decent hand-designed agent at the beginning. If your approach can also effectively improve upon that and is better than other baseline methods, the results will be more solid.
- The result will look even stronger if you can show your method even beat the previous works on continuous design optimization [2,3] (probably with attribute transform only and no skeleton transform).

Typos:
- JSMPLs -> JSMLPs in the caption of Figure 4.

References:
- [1] Gupta, Agrim, et al. "Embodied Intelligence via Learning and Evolution." arXiv preprint arXiv:2102.02202 (2021).
- [2] Luck, Kevin Sebastian, Heni Ben Amor, and Roberto Calandra. "Data-efficient co-adaptation of morphology and behaviour with deep reinforcement learning." Conference on Robot Learning. PMLR, 2020.
- [3] Schaff, Charles, et al. "Jointly learning to construct and control agents using deep reinforcement learning." 2019 International Conference on Robotics and Automation (ICRA). IEEE, 2019.

--------- Post Rebuttal Update ----------

The authors adequately addressed my main concerns through extra experiments and analysis.


**Summary Of The Paper:**

This paper proposes a transform-and-control policy to optimize the robotic agents' designs. Contributions include:
- A novel perspective on agent design: rather than formulating agent design as a bi-level optimization, this paper embeds both design generation and control into a single decision-making process such that both design and control are optimized by the same RL algorithm.
- In this formulation, the training experience from different designs is shared to improve sample efficiency.
- Joint-specialized MLP on top of the GNN policy that further finetunes the control of individual joints.

**Summary Of The Review:**

This paper provides an interesting perspective on efficient agent design with reasonable technical approaches, and the results look empirically good. I recommend acceptance.

---

> ### Author Response · Authors · 2021-11-22
> **Response to Reviewer mvHK [1/2]**
>
> Thank you for the constructive feedback and detailed review. We also appreciate the acknowledgment of our approach's novelty. We have revised the paper to include experiments with more seeds and baseline comparison of continuous design optimization for finetuning expert designs (please see the revision summary post for all changes). In the following, we aim to address your questions and concerns.
>
> ---
>
> > **Q1:** *"Experiments are only conducted on four custom environments..."*
>
> We’d like to mention that four environments are actually not few considering that NGE [1] has only used two environments. Our 2D Locomotion and Swimmer environments are also quite similar to the Walker and Fish environment in NGE despite using different morphology parametrization (we use capsules only). Furthermore, our 3D Locomotion and Gap Crosser environments cover other uses cases including 3D multi-legged robots and challenging terrains.
>
> ---
>
> > **Q2:** *"three random seeds are far below the standard"*
>
> We have rerun all the experiments with more seeds (6 seeds in total now) in the revised version. We hope the reviewer can understand that these new experiments cost a significant amount of compute, i.e., 100+ machine days in total, which limits the number of experiments we can run. We note that 6 seeds are also already more than many prior works in this area (e.g., 4 seeds in [2] and 3 seeds in [3]).
>
> ---
>
> > **Q3:** *"Discussing comparison with NGE (probably the strongest baseline) about similarities and differences and how those differences lead to a huge performance increase"
> > "Could you provide more intuition on the comparison against NGE?"*
>
> There are three main advantages of our method over NGE:
> 1. **NGE does not allow experience sharing among species in a generation.** As shown in Algorithm 1 of NGE [1], each species inside a generation $j$ has its own set of weights $\theta_i^j$ and is trained independently without sharing experiences among different species. The experience sharing in NGE is only enabled through weighting sharing between a species and its parent species from the previous generation. This means that if there are $N$ species in a generation, in every epoch, each species is only trained with $M/N$ number of experience samples where $M$ is the sample budget for each epoch. At the end of the training, each species has only used $EM/N$ samples for training where $E$ is the number of epochs. In contrast, in our method, every design shares the same control policy, so the policy can use all $EM$ samples. Therefore, our method allows better experience sharing across different designs, which improves sample efficiency.
> 2. **Our method uses a transform policy to change designs instead of random mutation.** Our transform policy takes the current design as input to output the transform actions (design changes). Through training, the policy learns to store information about which design to prefer and which design to avoid. This information is also shared among different joints via the use of GNNs, where joints in similar states choose similar transform actions (which is also balanced by JSMLPs for joint specialization). Additionally, the policy also allows every joint to simultaneously change its design (e.g., add a child joint, change joint attributes). For example, in 3D Locomotion, the agent can simultaneously grow its four feet in a single timestep, while ES-based methods such as NGE will take four different mutations to obtain four feet. Therefore, our method with the transform policy allows better generalization and experience sharing among joints, compared to ES-based methods that perform random mutation.
> 5. **Our method allows more exploration.** Our transform-and-control policy tries out a new design every episode, which means our policy can try $M/H_\text{avg}$ designs every epoch, where $M$ is the total number of sample timesteps and $H_\text{avg}$ is the average episode length. There is also more exploration for our approach at the start of the training when $H_\text{avg}$ is small. On the other hand, ES-based methods such as NGE only try $N$ (num of species) different designs every epoch. If NGE uses too many species (large $N$), each species will have few samples to train as mentioned in point 1. Therefore, $N$ is typically set to be $\ll M/H_\text{avg}$. For example, $N$ is set from 16 to 100 in NGE, while $M/H_\text{avg}$ in our method can be more than 2000.

---

> > ### Author Response · Authors · 2021-11-22
> > **Response to Reviewer mvHK [2/2]**
> >
> > > **Q4:** *"Enriching ablation studies by training with skeleton transformation disabled and attribute transformation disabled respectively."
> > > "...probably we already have a decent hand-designed agent at the beginning. If your approach can also effectively improve upon that..."
> > > "...show your method even beat the previous works on continuous design optimization..."*
> >
> > In Appendix A of the revised version, we have also evaluated our approach's ability to finetune a given expert design and compared it against a popular continuous design optimization baseline [4]. We perform experiments with the skeleton transform stage disabled in our approach and use the expert design as the initial design. As shown in Appendix A, our approach outperforms the expert design and baseline significantly for all four environments. This demonstrates that, in addition to combinatorial design optimization, our approach also enjoys superior performance in continuous design optimization.
> >
> > ---
> >
> > > **Q5:** *"I can't entirely agree with the argument that this approach enables first-order optimization of agent design. Technically, both this approach and ES-based methods do not have access to the ground-truth gradient and estimate first-order gradients based on the gathered experiences. So, this approach is still zeroth-order optimization, and it's not appropriate to claim that the sample efficiency comes from the first-order nature of the method."*
> >
> > We acknowledge that this statement is a bit confusing, and we have modified it in the revised version. What we intended to say by “first-order” is that our method uses a parametrized transform policy to output design-changing actions, which enables us to use policy gradients methods for policy optimization, and policy gradients use the first-order information in the policy by taking the exact gradient of the policy w.r.t. its parameters.
> >
> >
> > ---
> >
> > > **Q6:** *"While I agree that ES-based methods have a high-dimensional search space for design, your approach does not essentially reduce that search space..."*
> >
> > We intend to say that all methods (NGE and ours) face the same problem of high-dimensional design space, but our method, using a parametrized transform policy to output design changes, is more efficient because the policy allows better generalization and experience sharing among joints (as explained in the answer to Q3). We are not claiming that we reduce the design search space.
> >
> > ---
> >
> > > **Q7:** *"skeptical about the reported performance of NGE..."*
> >
> > We believe that our results, especially with more random seeds, are consistent with the findings in the NGE paper [1]. In particular, we find that NGE performs much better than RGS (1.5\~2x rewards) for 2D Locomotion, Swimmer, and 3D Locomotion. This agrees with the results in the NGE paper. Similar to prior work [5], we use our own implementation of NGE to have a fair comparison and avoid any performance differences caused by implementation details such as deep learning frameworks, RL algorithm implementation, etc.
> >
> > ---
> >
> > ### **Reference**
> >
> > [1] Wang et al., "Neural Graph Evolution: Towards Efficient Automatic Robot Design." ICLR 2019.
> > [2] Huang et al., "One Policy to Control Them All: Shared Modular Policies for Agent-Agnostic Control." ICML 2020.
> > [3] Kurin et al., "My Body Is A Cage: The Role of Morphology in Graph-based Incompatible Control." ICLR 2021.
> > [4] Ha, "Reinforcement Learning For Improving Agent Design." Artificial Life, 2019.
> > [5] Hejna et al., "Task-Agnostic Morphology Evolution." ICLR 2021.

---

> > > ### Comment · Reviewer_mvHK · 2021-11-29
> > > **Response to Authors**
> > >
> > > Thank you for the detailed clarifications. I think now it's ready for publication and I have updated my score to 8.
> > >
> > > However, I am still not satisfied with only 4 environments, which makes results overall less exciting. Also, I strongly encourage authors to incorporate more discussions on the limitations.

---

### Official Review · Reviewer_487L · 2021-11-02

**Correctness:** 3
**Technical Novelty And Significance:** 3
**Empirical Novelty And Significance:** 3
**Recommendation:** 8
**Confidence:** 4

**Main Review:**

Strengths:
- The resulting morphologies of the algorithm seem interesting and effective.
- The idea of breaking the training into multiple stages where policy first designs the agent and then controls it is novel.
- Paper is in general well written.

Weaknesses/Questions:
- Not sure how much control a user has over the design space. For example one may want the design to mimic certain animals, or be symmetrical.
- The training for the first two stages would involve a delayed sparse reward signal. Is there any intuition why PPO handles this okay in the presented case? Does the horizon of the first two stages, where no reward is given, impact the learning effectiveness?
- It’d be interested to analyze individual components of the proposed algorithm. E.g. comparing to [1] or [2] assuming a fixed morphology. This will help compare the condition policy approach to a bi-level optimization approach.

[1]. Reinforcement Learning for Improving Agent Design. Ha.
[2]. Jointly Learning to Construct and Control Agents using Deep Reinforcement Learning. Schaff et al.


**Summary Of The Paper:**

The paper introduced a reinforcement learning algorithm that simultaneously optimizes the design as well as the controller of a simulated robot to perform locomotion tasks. The core idea is to train a conditioned policy that performs the task in three stages: 1) morphology design of the robot, 2) design parameter adjustment, and 3) controlling of the robot to perform the task. By integrating the design process into the policy learning framework, they are able to design novel and effective agents to complete a variety of tasks. To support the proposed algorithm, graph neural networks is heavily used to support different morphologies. They further propose a joint-specific architecture to improve flexibility of the network, which improves the performance of the algorithm.

**Summary Of The Review:**

The paper presents a concrete algorithm with good results in general. It would be helpful to provide some more details and insights in what makes the method work and what are some limitations of the current work.

---

> ### Author Response · Authors · 2021-11-22
> **Response to Reviewer 487L**
>
> Thank you for the constructive feedback and appreciation of our approach's novelty. We have revised the paper to include experiments with more seeds and baseline comparison of continuous design optimization (please see the revision summary post for all changes). Here, we aim to address the questions raised in the review.
>
> ---
>
> > **Q1:** *"Not sure how much control a user has over the design space. For example one may want the design to mimic certain animals, or be symmetrical."*
>
> There are two ways one can control the design space in our method:
> 1. **Use design-related rewards in the transform stage.** For example, we can add rewards to encourage symmetry or certain topology in the design.
> 2. **Use certain design and action parametrization.** For example, to enforce symmetry, we can parametrize the design and transform actions with only one side of the agent's body, and if we change the design with transform actions, the other side of the agent's body is also symmetrically updated.
>
> ---
>
> > **Q2:** *"The training for the first two stages would involve a delayed sparse reward signal. Is there any intuition why PPO handles this okay in the presented case? Does the horizon of the first two stages, where no reward is given, impact the learning effectiveness?"*
>
> The main reason is that the first two transform stages are relatively short in our cases, i.e., only $N_\mathrm{s} + N_\mathrm{z} = 6$ timesteps, compared to the much longer horizon of the execution stage (> 100 timesteps). Therefore, the transform sub-policy can quickly receive reward signals after taking the transform actions.
>
> ---
>
> > **Q3:** *"It’d be interesting to analyze individual components of the proposed algorithm. E.g. comparing to [1] or [2] assuming a fixed morphology. This will help compare the condition policy approach to a bi-level optimization approach."*
>
> In Appendix A of the revised version, we have also evaluated our approach's ability to finetune a given expert design and compared it against a popular continuous design optimization baseline [1]. We perform experiments with the skeleton transform stage disabled in our approach and use the expert design as the initial design. As shown in Appendix A, our approach outperforms the expert design and baseline significantly for all four environments. This demonstrates that, in addition to combinatorial design optimization, our approach also enjoys superior performance in continuous design optimization.
>
> ---
>
> ### **Reference**
>
> [1] Ha, "Reinforcement Learning For Improving Agent Design." Artificial Life, 2019.

---

> > ### Comment · Reviewer_487L · 2021-11-30
> > **Response to author rebuttal**
> >
> > Thanks for the response!
> >
> > It's interesting to know that the design takes 6 steps in total only. I guess it makes sense since it can grow the robot exponentially. It'd be nice to make this clear explicitly in the paper.
> >
> > I'll keep my current score of 8.

---

### Author Response · Authors · 2021-11-22
**Revision Summary**

We thank the reviewers for their detailed and useful feedback. We are very grateful that all reviewers appreciate the novelty of our approach. We have revised the paper based on the feedback and comments. Here, we list the main changes in the revised paper (highlighted in blue):

1.  **More random seeds for all experiments.** We have increased the number of seeds to 6 for all the experiments in the revised paper.
2.  **Baseline comparison of continuous design optimization for finetuning expert designs**. We have added these experiments in *Appendix A* of the revised version. The results show that our approach outperforms the expert design and baseline significantly for all the environments.
3.  **Detailed discussions of our approach's advantages over ES-based methods such as NGE [1].** We have added these discussions in the introduction and *Appendix B* of the revised version as requested by Reviewer mvHK.
4.  **Discussion of attention-based policy [2]**. We have added a short discussion of [2] in the related work to address the comments of Reviewer 42PW.

If the reviewers still have remaining questions after reading the revision and responses, we are happy to provide additional responses.

---

### **Reference**

[1] Wang et al., "Neural Graph Evolution: Towards Efficient Automatic Robot Design." ICLR 2019.
[2] Kurin et al., "My Body Is A Cage: The Role of Morphology in Graph-based Incompatible Control." ICLR 2021.

---

### Decision · Program_Chairs · 2022-01-20

**Decision:**

Accept (Oral)

**Comment:**

The paper considers the problem of learning both the physical design (morphology and parameters) of a robot together with the corresponding control policy to optimize performance at a target task. Unlike several contemporary methods that formulate this as two separate, but coupled, optimization problems, the paper unifies these decisions into a single decision-making framework. More specifically, a conditional policy learns to first change an agent's physical design (i.e., the morphology/skeletal structure and its associated parameters), and then to control the design. The policy is formulated as a graph neural network, enabling a single policy to simultaneously control robots with different morphologies (and, in turn, different action spaces). Experimental results demonstrate that the approach outperforms recent baselines on a variety of simulated control tasks.

The paper considers an interesting and challenging problem, that of jointly optimizing an agent's physical design and its control policy, an area of research that has received renewed attention of-late. As the reviewers note, the idea of treating design and control in the context of a single decision-making process is novel. The approach is principled and the experimental results largely justify the significance of the contributions. The reviewers agree that the approach is described clearly and that the paper is well written. The reviewers initially raised a few concerns regarding the experimental evaluation, including the desire for more in-depth evaluations and the need for more random seeds. They also questioned some of the claims made in the initial submission. The authors provided a detailed response to each of these points and made changes to the paper to resolve most of the concerns.

In summary, the paper proposes a novel approach to an interesting problem with convincing results.